# OpenReview forum: "Efficient Synthetic Network Generation via Latent Embedding Reconstruction"
_ICML.cc/2026/Conference — ICML 2026 regular_

### Official Review · Reviewer_XJHc · 2026-02-22

**Soundness:** 3
**Presentation:** 3
**Significance:** 3
**Originality:** 2
**Overall Recommendation:** 4
**Confidence:** 3

**Summary:**

The paper proposes SyNGLER which is an efficient algorithm that could generate synthetic networks using latent space network models and node embeddings. In the first step, node embeddings are generated using a latent space model. Then, a generative model is trained to generate new latent embeddings and reconstruct the networks using the same latent space model. Experiments over simulated and real-world datasets are conducted to demonstrate its performance which shows similar or better results compared to several baselines while running more efficiently.

**Compliance With Llm Reviewing Policy:**

Affirmed.

**Final Justification:**

The author address my concerns, and I believe current score well represents the quality of the paper.

**Key Questions For Authors:**

**Questions**
1. It seems on some real-world datasets the proposed models cannot beat baselines (e.g. on DBLP). Why does this happen? What kinds of data distributions the proposed model may suffer compared to others? Is it because the real-world datasets do not necessarily follow the assumptions being made? Or is it due to the subgraph sampling being taken over the real-world dataset?
2. Although generation of attributed networks is discussed in a subsection and appendix, it is not comprehensive enough and thus limits the contribution of the paper since the majority of the real world graphs are attributed. Why not add attributed graphs generation as part of the main discussions / experiments in the paper?

**Limitations:**

No. The authors did not provide enough discussions for the limitations of the SyNGLER as well as its social impacts. More details should be discussed in the main paper.

**Strengths And Weaknesses:**

**Strength**
1. The paper addresses an important problem of generating synthetic networks preserving good network statistics efficiently compared to multiple baselines.
2. The paper considers multiple different generation scenarios from plain networks adjacency matrix generation to attributed networks generation. This aligns well with real-world graph datasets and problems.
3. Theoretical analysis is included to demonstrate the generation consistency of the proposed algorithm.
4. Extensive experiments have been conducted including synthetic datasets and real-world datasets. The visualization of generated graphs well reflects the good quality of generated graphs. An efficiency experiment is also conducted.

**Weakness**
1. Although the paper addresses the generation for attributed networks, the experiments are only conducted over link prediction tasks which is limited. The original ML utility experiment from GraphMaker paper also tests over node classification tasks which is also suitable for attributed networks generation evaluation which should be included as well.
2. It seems only VGAE is used as baselines over simulated networks. It would be better if the authors could provide experiment results for other more recent models such as GRAN / EDGE / GraphMaker over synthetic experiments as well.
3. Limitations of the proposed SyNGLER method are not clearly discussed: 1) It is not clear how the algorithm could be applied to larger real-world graphs given the experiments over real-world dataset only sample induced subgraphs from them. 2) What if the assumptions of the latent space model breaks? The appendix D studies this over synthetic data but large real-world graphs might not necessarily follow the same assumption made by authors. It is recommended to test against the real-world large dataset. Besides, the authors do not provide enough discussions of how the model performs on or be applied to more complex graph structures such as multiplex graphs etc.

---

> ### Author Rebuttal · Authors · 2026-03-31
>
> We appreciate the time and effort you devoted to reviewing our paper, and we thank you for recognizing the significance of the problem, the generality of the method, and our theoretical guarantees and extensive experiments, as well as for your insightful questions. Below, we address your questions one by one.
>
> **(Q1) Why does DBLP look harder for SyNGLER?**
>
> In DBLP, the extracted subgraph has extremely high clustering and strong degree assortativity, which are difficult to capture simultaneously with a low-dimensional latent space model. In SyNGLER, the degree parameters do not directly enforce high-degree-to-high-degree linking, so some local motif structure is harder to match.
>
> More broadly, this highlights that different methods are suited to different structural aspects and application goals: SyNGLER offers a  tradeoff between computational efficiency and overall stable performance, while some baselines may better fit certain local structures on specific datasets
>
> **(Q2+W1) Attributed graphs.** Thank you for your comments on attributed graphs. We agree that attributed graph generation is important. In this paper, we chose to first focus on graph generation itself so that the framework and its statistical interpretation can be presented clearly. Due to the space limits of the original submission, the attributed-graph component is currently brief, appearing mainly in Section 2.3 of the main text; in the revision, we are reallocating space to include more discussion of attributed graphs in the main paper.
>
> For the node classification task, we have conducted supplementary experiments on SyNGLER-Attr, as shown in the following table (close to 1 is considered good):
> | Method | GCN | MLP |
> |--------|-----|-----|
> | GraphMaker | 1.01 | 1.02 |
> | SyNGLER-Attr | 0.98 ± 0.02 | 0.98 ± 0.02 |
> This stated result confirms the effectiveness of SyNGLER-Attr in efficiently preserving both structural and attribute information needed for node classification.
>
> **(W2) Simulation baselines.** We agree that adding modern baselines such as GRAN, EDGE, and GraphMaker to the simulation would make the study more complete. Some of the main results are listed in the following table.
>
> | Method   | Tri RMSE | Tri Bias | Clus RMSE | Clus Bias |
> |----------|---------:|---------:|----------:|----------:|
> | SyNG-D*  | 4.60 | 4.37 | 15.7 | 15.2 |
> | SyNG-R*  | 5.37 | 5.11 | 18.1 | 17.7 |
> | VGAE*    | 9.70 | -9.16 | 53.1 | -50.2 |
> | GRAN     | 6.02 | -5.88 | 38.7 | -38.5 |
> | EDGE     | 2.80 | -2.75 | 25.5 | -25.4 |
>
> Because repeatedly training graph neural network baselines in this setting is costly, we have reduced the number of Monte Carlo replications to 20 for now and will continue to increase it. More broadly, these methods are computationally heavy, and in their original studies are typically trained and evaluated on real datasets only once, rather than repeatedly validated through Monte Carlo simulations requiring many retraining runs.
>
> Overall, SyNG-D and SyNG-R remain competitive across the metrics and substantially outperform VGAE. GRAN and EDGE perform well but are less stable on higher-order structural quantities such as the clustering coefficient. Meanwhile, SyNG-D/R offers a good tradeoff between **computational efficiency** and **stable overall performance**.
>
> **(W3-1) Larger graphs.** In the current real-data evaluation, we use **induced subgraphs** extracted from very large original graphs, mainly to avoid OOM issues for several baselines and to enable fair comparison across methods while still facilitating evaluation using the full graph.
>
> On the other hand, we do have a direct large-scale study in Appendix D on a **one-million-node SBM**, and Appendices C.8 and C.9 further provide scalability evidence for our method through wall-clock time, e-FLOPs comparisons, and complexity analysis.
>
> **(W3-2) Model Misspecification / more complex graph types.** Thank you for this important point.
>
> - Latent space models can be viewed as low-rank approximations to the logit of the underlying probability matrix. Therefore, even under model misspecification, they remain useful as flexible universal approximators when the latent dimension is chosen appropriately.
>
> - Empirically, our evaluation already probes misspecification: on real-world graphs, we assess not only degree and spectral statistics, but also higher-order structure through triangle density, clustering coefficient, and 4-graphlet frequency distance. We are also including more discussion on this aspect.
>
> - For more complex graph types such as multiplex or multilayer graphs, the current paper does not yet provide a dedicated model, but extensions of the latent-space framework to these settings can be developed naturally, and we will discuss them more clearly in the revision.
>
> Thank you again. We are continuing to run additional experiments and revise the manuscript to incorporate all of the above discussion, especially emphasizing our strengths and limitations.

---

> > ### Author Rebuttal · Reviewer_XJHc · 2026-04-01
> >
> > Thank you for the detailed rebuttal. Please remember to include all the additional details in the final version. I will maintain my scores.

---

> > > ### Author Response · Authors · 2026-04-04
> > >
> > > Thank you again for your time, helpful feedback, and support. We truly appreciate it.

---

### Official Review · Reviewer_XT99 · 2026-02-24

**Soundness:** 4
**Presentation:** 4
**Significance:** 3
**Originality:** 3
**Overall Recommendation:** 4
**Confidence:** 4

**Summary:**

This paper introduces SyNGLER, a framework for generating realistic synthetic networks by decoupling representation learning from generative modeling. It first learns low-dimensional node embeddings from an observed network using a likelihood-based latent space model and then reconstructs the latent space by training a distribution-free generator, such as a score-based diffusion model or a resampling sampler, on these embeddings. By operating in a low-dimensional space rather than the high-dimensional adjacency matrix space, SyNGLER significantly reduces computational overhead. The framework provides theoretical consistency guarantees and empirically outperforms baselines in preserving key structural properties like degree distributions, clustering coefficients, and spectral motifs across diverse real-world datasets. Furthermore, it maintains high machine learning utility for downstream tasks and scales effectively to networks with up to one million nodes.

**Compliance With Llm Reviewing Policy:**

Affirmed.

**Final Justification:**

The authors have addressed the concerns raised in my initial review through their rebuttal. I maintain a positive evaluation and recommend a Weak Accept (4).

**Key Questions For Authors:**

1. How does SyNGLER's score-based generative model specifically differ from the approach for hypergraphs described in Wu et al. (2025)? Given that both works utilize denoising diffused embeddings and similar discretization techniques for error analysis, could you elaborate on the unique architectural or algorithmic adaptations required to move from hypergraph structures back to general graphs?
2. In the simulated sparse network experiments (Table 1), the framework is only compared against VGAE. Given that modern models like GRAN and EDGE are evaluated on real-world data, why were they excluded from the simulated experiments? Providing these comparisons would be better.

**Limitations:**

yes

**Strengths And Weaknesses:**

Strengths
1. Theoretical Guarantees: The paper establishes a rigorous framework by decomposing KL divergence into three error terms and proving consistency and generalization results.
2. High Computational Efficiency: By operating in a low-dimensional latent space, SyNGLER trains in tens of seconds, significantly outperforming deep models like GRAN and EDGE which take hundreds or thousands of seconds.

Weaknesses
1. The paper shares a similar technical lineage with Wu et al. (2025) regarding denoising diffused embeddings. While SyNGLER focuses on general networks (graphs) and established latent space models, it lacks a detailed comparison to distinguish its specific innovations from Wu's work on hypergraphs.
2. In simulated sparse network experiments, the authors only compare against VGAE. Although they include modern models like GRAN, EDGE, and GraphMaker in real-world evaluations, adding these baselines to the controlled simulations would more robustly validate performance in the sparse regime.

---

> ### Author Rebuttal · Authors · 2026-03-31
>
> Thank you for taking the time and effort to review our paper. We appreciate your recognition of the method's high computational efficiency and theoretical guarantees, as well as your insightful questions. Below, we address your questions one by one.
>
> **(Q1 & W1) Relation to Wu et al. (2025).** Thank you for this important comment. We agree that part of our method is inspired by Wu et al. (2025), particularly its use of an embedding-based generation pipeline. However, graphs and hypergraphs differ substantially in both data format and underlying structure, so graph generation remains a distinct problem that warrants systematic study. Moving from hypergraphs to graphs fundamentally changes the statistical model, decoder, estimation procedure, and theory.
>
> More specifically, hypergraph generation models **set-valued higher-order incidences**, where the goal is to represent and generate multi-way interactions. By contrast, SyNGLER is designed for **pairwise edge generation**: each node has a latent representation $\phi_i=(z_i,\alpha_i)$, and edges are decoded through a graph-specific pairwise link function
> $P_{ij}=\sigma(z_i^\top z_j+\alpha_i+\alpha_j+\rho_n),$
> with node-level propensity terms and a global sparsity/intercept parameter $\rho_n$. This leads to a different first-stage statistical problem: SyNGLER must fit a **constrained latent space graph model** with graph-specific identifiability conditions and constraints, rather than being a special case of a hypergraph model.
>
> The algorithmic consequences are also different. In our setting, the generator is trained on fitted **node embeddings** from a graph latent space model, and the generated samples are mapped back to an $n\times n$ adjacency structure through pairwise edge probabilities. In the hypergraph setting, the decoder and sampling mechanism keep the node set fixed and instead reconstruct multi-way relations among the nodes, which are fundamentally different combinatorial structures over the same set of nodes. Thus, although both works use denoising/diffusion in latent space, the front-end statistical model and the back-end decoding and sampling pipeline are different.
>
> The same is true on the theory side. Our end-to-end analysis is graph-specific: Theorem 3.1 decomposes the network-level average KL gap as
> $n^{-2} d_{\mathrm{KL}}(P_A \,\|\, P_{\tilde A}^A)=E_\rho+E_\Phi+E_{\mathrm{gen}},$
> where $E_\rho$ is a graph sparsity/intercept term and $E_\Phi$ is the graph latent-embedding estimation term. This decomposition relies on the pairwise graph decoder and dyadic conditional independence, and is therefore specific to the graph setting rather than a direct restatement of a hypergraph result. In this sense, the overlap lies mainly in the use of a low-dimensional score-based generator, whereas the statistical model, generation mechanism, and end-to-end theory here are new and tailored to graphs.
>
> **(Q2) Sparse simulated experiments only compare with VGAE.** We agree that adding modern baselines such as GRAN, EDGE, and GraphMaker to the sparse simulated experiments would make the study more complete. Some of the main results are listed in the following table.
>
> | Method   | Tri RMSE | Tri Bias | Clus RMSE | Clus Bias |
> |----------|---------:|---------:|----------:|----------:|
> | SyNG-D*  | 4.60 | 4.37 | 15.7 | 15.2 |
> | SyNG-R*  | 5.37 | 5.11 | 18.1 | 17.7 |
> | VGAE*    | 9.70 | -9.16 | 53.1 | -50.2 |
> | GRAN     | 6.02 | -5.88 | 38.7 | -38.5 |
> | EDGE     | 2.80 | -2.75 | 25.5 | -25.4 |
>
> Because repeatedly training graph neural network baselines in this setting is computationally expensive, we have reduced the number of Monte Carlo replications to 20 for now and will continue to increase it. More broadly, these methods are computationally heavy, and in their original studies they are typically trained and evaluated on real datasets only once, rather than repeatedly validated through Monte Carlo simulations that require many retraining runs.
>
> Overall, SyNG-D and SyNG-R remain competitive across the metrics and substantially outperform VGAE. GRAN and EDGE perform well but are less stable on higher-order structural quantities such as the clustering coefficient. Meanwhile, SyNG-D/R offers a good **tradeoff** between **computational efficiency** and **stable overall performance**.
>
> Thank you again. We are continuing to run additional experiments and revise the manuscript to incorporate all of the above discussion.

---

> > ### Author Rebuttal · Reviewer_XT99 · 2026-04-02
> >
> > After reading the rebuttal, I have no further questions. Thank you.

---

> > > ### Author Response · Authors · 2026-04-04
> > >
> > > Thank you again for your time, helpful feedback, and support. We sincerely appreciate it.

---

### Official Review · Reviewer_a5Su · 2026-02-25

**Soundness:** 2
**Presentation:** 2
**Significance:** 2
**Originality:** 2
**Overall Recommendation:** 3
**Confidence:** 4

**Summary:**

This paper proposed SyNGLER, an efficient framework for large-scale synthetic graph generation. Instead of directly modeling the adjacency matrix, the method fitted a likelihood-based latent space model to extract low-dimensional node embeddings, then learned their continuous distribution using a generative model. Synthetic graphs were sampled from these embeddings under the assumption of conditional edge independence. Empirical results demonstrated that SyNGLER achieves competitive structural preservation and computational efficiency compared to existing baselines.

**Compliance With Llm Reviewing Policy:**

Affirmed.

**Final Justification:**

As some of my primary concerns have now been addressed, I have increased my score from 2 to 3 ("Weak Reject"). However, some nontrivial concerns stilll remain. The substantially larger gap between SyNG-D and SyNG-R on Yelp, unlike the negligible differences on PolBlogs, DBLP, and YouTube, warrants a clearer discussion, potentially in relation to dataset homophily/heterophily. Additionally, the metric notation could also be standardized (e.g., Deg., Clus., Orbit) for better readability and consistency with prior literature.

**Key Questions For Authors:**

1. Given that higher-order structures (e.g., orbits or 4-cycles) are not explicitly parameterized, is there any theoretical guarantee that such motif-level patterns can be faithfully preserved under this framework?

2. Although the proposed method operates in a node-level latent space, recent work such as Unifying Generation and Prediction on Graphs with Latent Graph Diffusion (NeurIPS 2024) adopts a graph-level latent representation for generative modeling. How does the proposed framework differ from such latent graph diffusion approaches, and can they be compared in practice?

**Limitations:**

Refer to weakness and questions.

**Strengths And Weaknesses:**

(S1) Scalability & Simplicity: By learning low-dimensional embeddings via likelihood-based modeling under conditional edge independence, SyNGLER avoids direct adjacency-matrix modeling and improves scalability.

(S2) Explicit Structural Modeling: The edge formulation combines embedding inner products with node-specific degree terms, explicitly separating similarity and propensity effects beyond standard inner-product decoders.

(S3) Theoretical Guarantees: The paper provides KL-based error decomposition and consistency results under a sparse latent space model, showing that generation error diminishes under suitable conditions.

(W1) Readability: Eq. (2) presents the constraint set of the optimization problem in a dense inline form, e.g., (Z,α,ρ)∈R^(n×r)×R^n×R:  Z^⊤ 1_n=0_r,  α^⊤ 1_n=0,  max_i ||(z_i,α_i)||_2≤R, which reduces readability despite these identifiability and boundedness conditions being standard.

(W2) Notation and Presentation of ρ_n: Although ρ_n is treated as a global intercept jointly optimized with Z and α, its precise interpretation and implementation details were not fully specified in the main text. Since key clarifications were deferred to the appendix, the formulation is not entirely self-contained and may reduce clarity for readers.

(W3) Necessity of the Two-Stage Design for Adapting Other Generative Models: While the framework enables flexibility by allowing different generative models in the latent space, this design inherently requires two separate training stages, making end-to-end optimization difficult.

(W4) Limited Expressiveness for Higher-Order Graph Structures: The edge probability formulation π_ij=z_i^⊤ z_j+α_i+α_j+ρ_n models pairwise similarity and node-level propensity but does not explicitly parameterize higher-order structures such as motifs (e.g., orbits or 4-cycles), which must be captured indirectly through latent embeddings.

(W5) Missing Comparisons with Recent Scalable State-of-the-Art Baselines: Although efficiency and scalability were emphasized as key contributions, the evaluation omits recent scalable graph generators such as Iterative Local Expansion for Scalable Graph Generation (ICLR 2024) and Hierarchical Graph Generative Networks (HiGen) (ICLR 2024). Including comparisons on structural fidelity and computational efficiency would strengthen the rigor of the empirical assessment.

---

> ### Author Rebuttal · Authors · 2026-03-31
>
> We appreciate the time and effort you devoted to reviewing our paper, and thank you for recognizing the scalability, explicit structural modeling, and theoretical guarantees of our method, as well as your thoughtful questions and critiques. Below, we respond point by point.
>
> **(Q1+W4) Higher-order structure.** Thank you for this critique. Our method can capture higher-order graph structure both theoretically and empirically.
>
> - Theoretically, expected fixed-order motifs are functionals of the edge-probability matrix $P$. For example, expected triangles
>   $t_\triangle(P)=\binom{n}{3}^{-1}\sum_{i<j<k} P_{ij}P_{ik}P_{jk}$.
>   Since embeddings consistently recover $P$, the corresponding motif densities are also consistently recovered by plug-in. The same idea applies to other fixed-size motifs such as 4-cycles.
> - Empirically, our evaluation already reflects this: we assess similarity in higher-order metrics such as triangle density, clustering coefficient, and graphlet statistics.
>
> In short, although motifs are not explicitly parameterized, they can still be captured both theoretically and empirically. We will make this more explicit in the revision by adding a formal result for fixed-order motif statistics.
>
> **(Q2) Latent graph diffusion.** Thank you for pointing out this reference. Although both LGD and our method use latent-space generation, they differ fundamentally.
>
> - **Task.** LGD targets graph-level generation for multi-graph datasets such as molecules, whereas our method generates networks with similar structure from a single large observed graph.
> - **Representation.** LGD learns graph-level embeddings through an encoder-decoder architecture, while our method fits a likelihood-based latent space model and obtains interpretable node-level embeddings.
> - **Scalability.** LGD relies on dense $O(N^2)$ edge features and is not designed for large sparse networks, whereas our framework is.
>
> We will revise the manuscript to clarify these differences more explicitly.
>
> **(W1).** Thank you for your comment on the dense presentation of the constraint set in Eq. (2). The constraint set is included for technical rigor. Following your suggestion, we moved its definition outside the equation and revised the related expressions to improve readability.
>
> **(W2)** Thank you for this suggestion. In the revision, we added a clarification of $\rho$ in the main text to make the formulation more self-contained and improve the presentation of this notation.
>
> **(W3) Two-stage design.** We agree that our method does not use fully end-to-end joint optimization, and this separation is intentional.
>
> First, end-to-end network generation is computationally demanding, especially for large sparse graphs, since it typically handles discrete adjacency structure directly or relies on iterative variational approaches. Our two-stage design avoids this burden and is therefore computationally efficient.
>
> Second, the embedding stage is based on a latent space model, which provides interpretable node embeddings and a well-understood foundation. This separation also allows flexible choices of `GenModel` and yields the interpretable error decomposition in Theorem 3.1, where $E_{\mathrm{gen}}$ isolates the latent-generation error.
>
> In short, the two-stage design is chosen to improve computational efficiency, interpretability, and flexibility.
>
> **(W5) Additional baselines.** We appreciate the suggestion regarding additional recent baselines. We conducted additional empirical studies on these baselines. Some of the results are shown in Tables 1 and 2.
>
> **Table 1:** Tri./Clus. (lower is better)
>
> | Method | PolBlogs | DBLP | YouTube | Yelp |
> |---|---:|---:|---:|---:|
> | SyNG-D | 0.7/1.9 | 1.6/6.2 | 1.0/2.3 | 2.0/1.8 |
> | SyNG-R | 0.8/2.5 | 1.4/3.8 | 1.1/2.2 | 0.8/0.8 |
> | HiGen | 40/3.8 | 1590/49 | 411/30 | — |
> | ILDG | 0.02/14 | 155/34 | OOM | OOM |
>
> **Table 2:** Eig./DegC MMD
>
> | Method | PolBlogs | DBLP | YouTube | Yelp |
> |---|---:|---:|---:|---:|
> | SyNG-D | 2.6/1.0 | 8.8/9.3 | 2.6/2.1 | 2.5/6.7 |
> | SyNG-R | 3.0/0.9 | 8.0/7.5 | 4.5/1.1 | 4.7/0.6 |
> | HiGen | 3.5/0.2 | 0.8/0.01 | 16/0.05 | — |
> | ILDG | 3.2/0.00 | 5.9/1.6 | OOM | OOM |
>
> - HiGen runs on all four datasets and shows reasonable spectral performance, but its triangle density and clustering are less accurate, and it is also relatively time-consuming on large datasets (about 24 hours on Yelp).
> - ILDG encounters memory limits beyond 1,500 nodes; on smaller graphs, its expansion procedure can terminate early and result in smaller graphs.
>
> We also appreciate the reviewer for pointing out these two relevant papers. We now include them in the literature review and are revising the manuscript accordingly. Overall, our method is better suited for **large-scale sparse network generation**, whereas these baselines are designed for different settings.
>
> Thank you again. We are continuing to run additional experiments and revise the manuscript to incorporate all of the above discussion.

---

> > ### Author Rebuttal · Reviewer_a5Su · 2026-04-02
> >
> > - Q2, W1, W2: Well addressed.
> >
> > - W3: The authors’ explanation is reasonable, and stating this point more
> > explicitly as a limitation or future direction in the conclusion would further
> > improve the paper.
> >
> > - W5 (partially addressed): The additional experiments with the suggested SOTA methods are appropriate, and the provided results and interpretations appear reasonable.
> >
> > - Q1+W4. Higher-order structure (unresolved): The response does not yet fully resolve this concern. The current discussion still relies mainly on triangle- and clustering-based statistics, which provide only limited evidence for broader higher-order structural fidelity. In prior graph generation work, including the already cited GRAN and EDGE, orbit-based metrics are used to evaluate higher-order motif structure, and this has become a standard practice in the literature. Including such results in the main comparison tables would therefore provide a more complete and convincing validation of this claim.
> >
> > - W5. baseline positioning (unresolved): As the authors note that their method is particularly suited for large-scale sparse network generation, it may also be helpful to compare against methods specifically designed for large single-graph generation. In particular, CELL (NetGAN without GAN) seems highly relevant in this setting and may serve as a more natural baseline than VGAE. Since its random-walk formulation and stationary distribution π encode global degree-related structure, it is closely connected to the type of structural information emphasized in this paper. It may therefore be worthwhile to
> > consider including CELL as an additional baseline.

---

> > > ### Author Response · Authors · 2026-04-04
> > >
> > > We thank you for carefully reading our rebuttal and for acknowledging that part of the concerns have been addressed. We also appreciate your further comments on the higher-order structure and baseline positioning. In response, we have conducted additional empirical studies and included further discussion of CELL.
> > >
> > > - **Q1+W4. Higher-order structure.** We thank the reviewer for this suggestion. We have added orbit-based metrics (Orbit, using 4-node graphlet MMD, and 4-Cyc, 4-cycle density) to our evaluation, following the protocol of GRAN (Liao et al., 2019). We also include CELL (Rendsburg et al., 2020) as an additional baseline, as suggested in W5. Part of the main results are summarized below (lower values indicate better performance).
> > >
> > > **PolBlogs** (n=1,222)
> > >
> > > | Method | Tri | Clus | Eig | DegC | Orbit | 4-Cyc |
> > > |---|---:|---:|---:|---:|---:|---:|
> > > | SyNG-D | 0.71 | 1.90 | 2.58 | 0.97 | 0.0001 | 0.14 |
> > > | SyNG-R | 0.83 | 2.45 | 3.01 | 0.92 | 0.0002 | 0.16 |
> > > | CELL | 1.00 | 5.14 | 0.00 | 0.003 | 0.005 | 0.20 |
> > > | EDGE | 0.79 | 5.69 | 4.69 | 0.00 | 0.043 | 2.88 |
> > > | GRAN | 1.57 | 11.4 | 9.04 | 13.9 | 0.290 | 5.11 |
> > >
> > > **DBLP** (n=1,481)
> > >
> > > | Method | Tri | Clus | Eig | DegC | Orbit | 4-Cyc |
> > > |---|---:|---:|---:|---:|---:|---:|
> > > | SyNG-D | 1.58 | 0.62 | 0.88 | 0.93 | 0.035 | 0.45 |
> > > | SyNG-R | 1.43 | 0.38 | 0.80 | 0.75 | 0.009 | 0.41 |
> > > | CELL | 4.68 | 3.20 | 2.31 | 3.35 | 0.099 | 1.14 |
> > > | EDGE | 2.20 | 1.33 | 1.39 | 0.99 | 0.347 | 237 |
> > > | GRAN | 7.99 | 8.92 | 1.76 | 2.97 | 1.425 | 135 |
> > >
> > > **YouTube** (n=1,991)
> > >
> > > | Method | Tri | Clus | Eig | DegC | Orbit | 4-Cyc |
> > > |---|---:|---:|---:|---:|---:|---:|
> > > | SyNG-D | 0.97 | 2.28 | 2.61 | 2.13 | 0.002 | 0.09 |
> > > | SyNG-R | 1.11 | 2.23 | 4.47 | 1.10 | 0.0003 | 0.05 |
> > > | CELL | 0.62 | 3.40 | 2.17 | 0.00 | 0.004 | 0.09 |
> > > | EDGE | 0.79 | 14.4 | 4.44 | 7.84 | 0.512 | 598 |
> > > | GRAN | 1.13 | 10.1 | 7.88 | 7.86 | 0.577 | 17.3 |
> > >
> > > **Yelp** (n=4,530; GRAN/EDGE OOM)
> > >
> > > | Method | Tri | Clus | Eig | DegC | Orbit | 4-Cyc |
> > > |---|---:|---:|---:|---:|---:|---:|
> > > | SyNG-D | 2.00 | 1.77 | 2.50 | 6.72 | 0.0003 | 0.21 |
> > > | SyNG-R | 0.78 | 0.76 | 4.69 | 0.62 | 0.0004 | 0.28 |
> > > | CELL | 2.67 | 4.44 | 5.67 | 2.96 | 0.004 | 0.65 |
> > >
> > > On the orbit-based and 4-cycle metrics, SyNGLER achieves much lower error than GRAN and EDGE, confirming that the latent-space approach preserves higher-order structural properties beyond triangles and clustering. CELL, which preserves the degree sequence by construction, also performs reasonably on some orbit metrics, suggesting that degree preservation may help retain certain local structural patterns, though it does not fully capture triangle density or clustering coefficient. On Yelp ($n=4{,}530$), both EDGE and GRAN run out of memory, while SyNGLER and CELL scale without difficulty.
> > >
> > > - **W5. CELL baseline.** We agree that CELL is a natural baseline for this setting. As shown above, CELL preserves local degree structure well, but performs relatively worse than SyNGLER on triangle density, clustering coefficient, and overall spectral fidelity.
> > >   - CELL is based on a low-rank approximation of the edge-probability matrix (before the softmax transformation) and does not explicitly separate the roles of different latent factors in edge formation. In contrast, SyNGLER decomposes edge formation into latent node position ($z$), node degree heterogeneity ($\alpha$), and global sparsity ($\rho$), yielding a more structured and interpretable model.
> > >   - With this decomposition, we can study the identifiability of the embeddings, which gives SyNGLER both interpretability and theoretical support. It also enables more principled model diagnosis, such as uncertainty quantification and model selection, in a way that is grounded in the latent space literature.
> > >   - Moreover, SyNGLER learns a generative distribution over node embeddings, allowing it to generate new nodes in the synthetic networks. CELL, by contrast, samples graphs from a fixed learned edge-probability matrix. This makes SyNGLER especially appealing for applications such as synthetic data augmentation, where structural diversity across generated graphs is important and where new nodes should be produced to avoid directly reusing information from the original nodes.
> > >
> > > We will include the corresponding discussion and additional results in the paper. Thank you again for your helpful comments.

---

### Official Review · Reviewer_tqCh · 2026-03-12

**Soundness:** 3
**Presentation:** 3
**Significance:** 3
**Originality:** 3
**Overall Recommendation:** 4
**Confidence:** 4

**Summary:**

This paper proposes SyNGLER, a two-stage framework for efficient synthetic network generation. The method first fits a likelihood-based latent space network model to obtain low-dimensional node embeddings (including a degree-related component to capture heterogeneity and a global sparsity parameter), and then learns a generator in the embedding space (either resampling or a score-based diffusion model) to produce new latent embeddings. Synthetic graphs are generated by sampling edges conditionally independent given the generated embeddings under the fitted latent model. The paper develops a KL-based error decomposition that separates sparsity-parameter error, embedding-estimation error, and embedding-generator error, and provides consistency-style analyses for the first two terms under sparse regimes. Experiments on simulated and real networks evaluate preservation of network statistics, ML utility (train on synthetic, test on real), and computational efficiency (e-FLOPs), showing favorable performance and strong efficiency compared to several existing baselines.

**Compliance With Llm Reviewing Policy:**

Affirmed.

**Final Justification:**

The rebuttal addressed our concerns, and the proposed SyNGLER framework is sound and original, effectively combining latent space modeling with modern generative techniques for synthetic network generation. We maintain a positive evaluation and the score of 4 (weak accept).

**Key Questions For Authors:**

(1) Robustness to latent-model misspecification: Can you include (or at least discuss) a stress test where the observed graph exhibits pronounced higher-order dependence that is difficult to capture by your latent space model with conditionally independent edges (e.g., strong triadic closure/motif effects beyond what the fitted embedding model explains)? This would help clarify the boundary of applicability and whether SyNGLER degrades gracefully under misspecification.

(2) End-to-end theoretical implication: Theorem 3.1 decomposes the average KL gap into terms $E_\rho + E_\Phi + E_{\mathrm{gen}}$. The analysis focuses on $E_\rho$ and $E_\Phi$. Could you explicitly state what assumptions on the embedding generator (e.g., the diffusion model trained on $\hat\Phi$) would ensure $E_{\mathrm{gen}} \to 0$ at an explicit rate, and how the required conditions scale with embedding dimension and sample size (number of nodes)?

(3) Baseline protocol and fairness: Appendix C.4 states that for VGAE/EDGE/GRAN you use the default models from the corresponding codebases without further tuning. Since some baselines report OOM on larger graphs and efficiency is a key claim, can you clarify (i) whether all methods are compared under comparable tuning/sampling budgets, and (ii) how e-FLOPs are computed (what components are counted for each method, e.g., training only vs training+sampling, and whether latent-model fitting time is included for SyNGLER)?

(4) Resampling vs diffusion in practice: The paper offers both resampling and diffusion-based embedding generation. When should a practitioner prefer resampling versus diffusion? In particular, does resampling mainly reproduce the empirical embedding distribution (more like bootstrap), and how do novelty/diversity and privacy-type concerns differ between the two modes?

(5) Reproducibility and code release: The appendix provides hyperparameters and environments, which is helpful. Do you plan to release code and evaluation scripts for SyNGLER (including the metric computation and efficiency accounting) to facilitate independent replication and fair comparison under the same pipeline?

**Limitations:**

yes

**Strengths And Weaknesses:**

Strengths:

(1) The overall idea is clean and practically motivated: moving generation from the adjacency space to a low-dimensional embedding space can substantially reduce training and sampling costs while preserving key structural characteristics.

(2) The latent model explicitly addresses sparsity and degree heterogeneity, which are central in real networks.

(3) The evaluation is broad and aligned with the stated goals: structural statistics, distributional distances, ML utility ratios, and efficiency comparisons.

(4) The theoretical organization via a modular KL decomposition is sensible and helps clarify which parts of the pipeline drive approximation error.

Weaknesses:

(1) The generator inherits a ceiling from the latent space model: if the fitted latent model is misspecified for a given network (e.g., strong higher-order dependence, motif-driven effects, or edge correlations not captured by conditional independence given embeddings), it is unclear how well SyNGLER can preserve such structure beyond what the latent model can represent.

(2) The theoretical results bound the sparsity-parameter and embedding-estimation terms, while the embedding-generator term is left as a modular component to be handled by generic diffusion theory. This is reasonable, but the end-to-end implication would be clearer if the paper stated concrete conditions/rates for the generator term sufficient to make the overall KL gap vanish.

(3) Baseline comparability and reproducibility remain potential concerns. The appendix provides substantial implementation detail, but the baseline protocol relies on external codebases and ``default models'' for some baselines, and there are OOM outcomes for certain methods on larger graphs. These choices can materially affect both quality and efficiency comparisons.

---

> ### Author Rebuttal · Authors · 2026-03-31
>
> Thank you for taking the time and effort to review our paper. We appreciate your recognition of the paper’s clarity, motivation, theoretical results, and broad evaluation, as well as your insightful questions. Below, we address your questions one by one.
>
> **(W1 & Q1) Robustness to latent-model misspecification / higher-order dependence.** The ability to handle model misspecification and to capture higher-order dependence are indeed important. Below, we explain why our model remains practically useful under such scenarios:
>
> - Latent space models may be viewed as low-rank approximations to the logit of the underlying probability matrix. With an appropriate latent dimension, the LSM can approximate any true probability matrix well. Since our first stage consistently recovers $P$, certain motifs and other structures that are functionals of $P$ can also be recovered consistently. Moreover, when edges are correlated, our approach can be interpreted as a consistent pseudo-likelihood method, as justified by Li et al. (2023).
>
> - Empirically, our evaluation already considers higher-order dependence and model misspecification. On real-world graphs, we assess not only degree and spectral statistics, but also higher-order structure through triangle density, clustering coefficient, and 4-graphlet frequency distance. Since these datasets are not assumed to follow our sparse latent-space model exactly, they already serve as a stress test beyond the idealized setting.
>
> We agree that an additional targeted misspecification experiment with stronger motif-driven dependence would further clarify the method’s graceful degradation, and we will add this stress test to the paper.
>
> **(W2 & Q2) End-to-end implication of Theorem 3.1 / explicit generator conditions.** Thank you for this helpful suggestion. The current theory provides an error decomposition and treats the generator as a flexible module that learns the distribution of the fitted embeddings.
>
> - The latent generation error $E_{\mathrm{gen}}$ converges to zero for many common generators, including the score-based diffusion model used in this paper, especially since the embedding dimension is moderate.
>
> - We will add results for this term under score-based diffusion models and state conditions and architectures under which $E_{\mathrm{gen}} \to 0$ at an explicit rate, with the embedding dimension and sample size incorporated explicitly.
>
> **(W3 & Q3) Baseline protocol / fairness / e-FLOPs accounting.** We appreciate this concern and will clarify the protocol more explicitly.
>
> - We compare methods under the same stated compute environment and report how the computing consumption scales with graph size. OOM cases are therefore essentially part of the reported scalability.
>
> - In Section 4.3, e-FLOPs measure training cost only. For neural baselines, we use forward-pass operation counts as a hardware-agnostic proxy; for the tree-based part of SyNG-D, we use calibrated tree-operation counts. Thus, e-FLOPs are a unified operation-count proxy rather than exact hardware-level FLOPs. For fairness, we fix the latent dimension across methods. OOM entries reflect scalability under the stated hardware budget. Latent fitting in SyNG-D is very much faster than diffusion training and is not included in the current report; we will add results including it.
>
> - For wall-clock time, Appendix C.8 reports training and sampling separately: each method is trained under its stated/default schedule, and then 128 graphs are sampled to measure average sampling time.
>
> **(Q4) Resampling vs diffusion in practice.** These two modes serve different purposes. Resampling is intended for settings where the generated network should incorporate existing nodes from the observed network, while diffusion is intended for generating new nodes through new latent embeddings.
>
> - **Application.** Resampling retains existing nodes, whereas diffusion generates new nodes; the two can also be combined.
> - **Novelty.** Diffusion is typically more diverse and novel, and is intuitively less tied to individual embeddings.
> - **Privacy.** Our method does not provide a formal privacy guarantee. Diffusion only avoids direct replication; rigorous privacy protection could be built on top of our method using, for example, differential privacy.
>
> **(Q5) Reproducibility / code release.** Thank you for mentioning this and we fully agree that reproducibility is important. Current appendix already contains implementation detail, including hyperparameters, preprocessing, evaluation metrics, baseline codebases, device environment and the efficiency-accounting protocol. We have planed to release the SyNGLER code, evaluation scripts, and efficiency-accounting pipeline upon acceptance / at camera-ready.
>
> Thank you again, and we are currently revising the manuscript to incorporate all of the above discussion.
>
> Reference:
> - Li, Jinming, et al. "Statistical inference on latent space models for network data." *arXiv preprint arXiv:2312.06605* (2023).

---

> > ### Author Rebuttal · Reviewer_tqCh · 2026-04-02
> >
> > I thank the authors for addressing the concerns well; I have no further comments.

---

> > > ### Author Response · Authors · 2026-04-04
> > >
> > > Thank you again for your time, helpful comments, and support; we sincerely appreciate it.

---

### Decision · Program_Chairs · 2026-04-30

**Decision:**

Accept (regular)

**Comment:**

This paper proposes SyNGLER, a framework for generating synthetic graphs. During training, SyNGLER learns low-dimensional latent node representations using a latent space network model. It then fits a distribution-free generator to model the distribution of these embeddings. For generation, SyNGLER samples node representations from this generator and reconstructs synthetic graphs from them via the latent space model.

The paper received mixed reviews with three reviewers in favor and one against. The reviewers agreed that the proposed framework is both scalable and theoretically well grounded, and that the experimental evaluation is thorough. However, they raised several serious concerns regarding the empirical evaluation. Some of these concerns (e.g., the use of only VGAE as a baseline on synthetic datasets) were addressed in the rebuttal, but others still remain. In particular, the evaluation is limited to small-scale datasets, and it is not clear whether the reported results are fully fair since the baselines appear to rely on default configurations from their respective codebases without additional tuning. Moreover, concerns were raised about the gap in the performance between SyNGLER-D and SyNGLER-R on Yelp. Another concern raised in the reviews relates to the expressive power of the latent model and whether it can capture higher-order structural patterns.

I believe that synthetic graph generation is an important research direction, and that this paper can constitute an addition to the body of work addressing this problem. The proposed framework is efficient and I appreciate the theoretical analysis provided. I also agree with the reviewers that the empirical evaluation is extensive. Regarding the fairness of the comparisons, Appendix C.4 suggests that some hyperparameter tuning is performed, at least for VGAE and GRAN. Moreover, the authors attribute the performance gap between SyNGLER-D and SyNGLER-R on Yelp to dataset scale. With respect to concerns about the expressive power of the latent model, the authors argue that motifs and other structures that are functionals of the edge-probability matrix can be recovered consistently, and they also include orbit-based metrics to support this claim. In my view, the main weakness of the paper is that the framework is not evaluated on truly large-scale datasets. While some of the considered datasets are large, only the largest connected components are extracted, resulting in relatively small graphs. Since efficiency is the main selling point of the paper, one would expect evaluation on large-scale real-world datasets, which is currently missing.

After carefully weighing the strengths and weaknesses, I view this as a borderline submission, but slightly on the acceptance side. The authors are encouraged to carefully address and incorporate all reviewer feedback in the revised version of the paper.